# Incremental False Negative Detection for Contrastive Learning

**Tsai-Shien Chen**[1]**, Wei-Chih Hung**[2]**, Hung-Yu Tseng**[3]**, Shao-Yi Chien**[1]**, Ming-Hsuan Yang**[3,4,5]

[1]National Taiwan University
[2]Waymo LLC
[3]University of California, Merced
[4]Yonsei University
[5]Google Research
tschen@media.ee.ntu.edu.tw

## Abstract

Self-supervised learning has recently shown great potential in vision tasks through contrastive learning, which aims to discriminate each image, or instance, in the dataset. However, such instance-level learning ignores the semantic relationship among instances and sometimes undesirably repels the anchor from the semantically similar samples, termed as "false negatives". In this work, we show that the unfavorable effect from false negatives is more significant for the large-scale datasets with more semantic concepts. To address the issue, we propose a novel self-supervised contrastive learning framework that incrementally detects and explicitly removes the false negative samples. Specifically, following the training process, our method dynamically detects increasing high-quality false negatives considering that the encoder gradually improves and the embedding space becomes more semantically structural. Next, we discuss two strategies to explicitly remove the detected false negatives during contrastive learning. Extensive experiments show that our framework outperforms other self-supervised contrastive learning methods on multiple benchmarks in a limited resource setup. The source code is available at https://github.com/tsaishien-chen/IFND.

## 1 Introduction

Self-supervised learning of visual representation (Doersch et al., 2015; Pathak et al., 2016; Noroozi & Favaro, 2016; Oord et al., 2018; Gidaris et al., 2018; Chen et al., 2020a) aims to learn a semantic-aware embedding space based on the image data without the supervision of human-labeled annotations. Recently, significant advances have been made by contrastive learning approaches (Hjelm et al., 2019; Tian et al., 2019; Chen et al., 2020b;c; He et al., 2020; Chen et al., 2020d) to reduce the performance gap with the supervised counterparts. Most unsupervised contrastive learning methods are developed based on the instance discrimination assumption. These approaches treat each training image as an individual instance and learn the representations that discriminate every single sample. Specifically, considering an arbitrary image as an anchor, the only positive sample is generated by applying a different data augmentation to the anchor image, while all other images are treated as negative samples. The training objective is to attract the positive to the anchor while repelling the negative samples from the anchor.

While instance-level contrastive learning has shown impressive performance, these methods do not take the semantic relationship among images into consideration. Although some images share similar semantic concepts with the anchor, they are still considered as negative samples and are equally pushed away from the anchor with all other negatives. We define these samples as "false negatives" in self-supervised contrastive learning, which would adversely affect the representation learning (Saunshi et al., 2019). One might argue that such undesirable effects might be minor for the datasets with diverse semantic concepts since the probabilities of drawing false negatives are relatively low. However, as shown in Figure 1, we empirically find the opposite results that the performance drops due to training with false negative samples are more significant on large-scale datasets with more semantic categories. These results bring up the issue of instance-level contrastive learning when it is applied to datasets with more complex semantic contents.

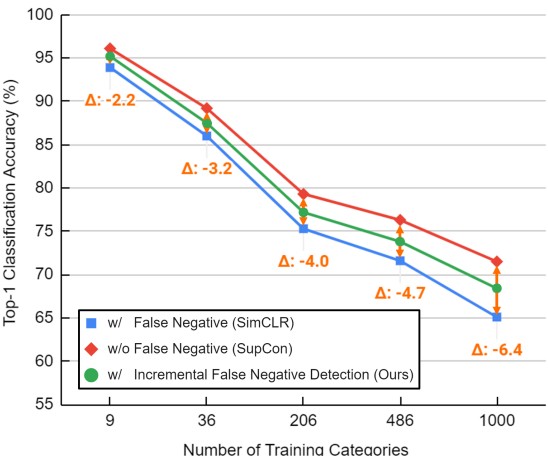

Figure 1: **Effect of false negative for contrastive learning.** To study the effect of false negatives, we compare two frameworks: Sim-CLR (Chen et al. (2020b), blue) representing instance-level contrastive learning which is trained with false negatives, and SupCon (Khosla et al. (2020), red) benefiting from human-labeled annotations to exclude false negatives. We observe that training with false negatives leads to larger performance gaps (orange) on the datasets with more semantic categories. The proposed approach (green) effectively alleviates such adverse effects by explicitly removing the detected false negatives for self-supervised contrastive learning. See Section 4.3 for experimental details.

To handle the false negative issue, we propose a novel framework to *incrementally detect and explicitly remove false negatives* for self-supervised contrastive learning. During training, we cluster samples in the embedding space and assign the cluster indices as the pseudo labels to the training images. Instances with an identical label to the anchor are then detected as false negatives. However, we notice that the pseudo labels generated in the earlier training stages are comparably unreliable to represent the semantic relationship among instances since the semantic structure of the embedding space is still under development. Therefore, we propose a novel strategy to assign the pseudo labels in an incremental manner. In the early training stage, we only use a small portion of pseudo labels with higher confidence while treating other samples as individual instances. In the later training stage, as the pseudo labels become more reliable, we dynamically include more confident labels to detect and remove false negatives and hence, can benefit semantic-aware representation learning.

Furthermore, we discuss two types of training objectives to explicitly address the detected false negatives during contrastive learning: elimination and attraction. We theoretically show that they are the generalization of triplet loss (Schroff et al., 2015). While both objectives avoid discriminating the anchor and its false negatives, the attraction objective further minimizes the distances between them. Empirically, we find that such a more aggressive strategy is less tolerant to noisy pseudo labels and unsuitable to our self-supervised framework. The main contributions of this paper are summarized as follows:

- We highlight the issue of false negatives in contrastive learning, especially on the large-scale datasets with more diverse semantic contents.
- We propose a novel contrastive learning framework with incremental false negative detection (IFND) that smoothly bridges instance-level and semantic-aware contrastive learning. Our approach explicitly detects and incrementally removes false negatives as the embedding space becomes more semantically structural through the training process.
- Our approach performs favorably against the existing self-supervised contrastive learning frameworks on multiple benchmarks under a limited training resource setup. Besides, we introduce two metrics: mean true positive/negative rates to evaluate the clustering quality.

## 2 RELATED WORK

**Instance-level contrastive learning.** Learning semantic-aware visual representation without human supervision is an essential but challenging task. Numerous approaches are developed based on the instance discrimination task (Dosovitskiy et al., 2015; Wu et al., 2018; Ye et al., 2019; Misra & Maaten, 2020), where each instance in the dataset is treated as an individual sample and a representation is learned to classify or contrastively separate samples. Recently, with the contrastive loss (Gutmann & Hyvärinen, 2010) and a set of image transformations or augmentations (Chen et al., 2020b), the instance-level contrastive learning methods (Hjelm et al., 2019; Tian et al., 2019; Chen et al., 2020b;c; He et al., 2020; Chen et al., 2020d) achieve state-of-the-art results and even outperform some supervised pre-training approaches. While significant improvement is achieved, instance-level contrastive learning frameworks typically do not take high-level semantics into ac-

count. For example, any two samples in the dataset would be considered as a negative pair and be pushed apart by the contrastive loss although some of them are semantically similar and should be nearby in the embedding space. Especially when the huge amounts of negative samples (8192 for Chen et al. (2020b), 65536 for He et al. (2020)) play a critical role in the success of contrastive learning, these instance-level approaches inevitably draw some "false negatives" and learn less effective representation models as shown by a recent study (Saunshi et al., 2019). In this paper, we further show that the performance gap between the contrastive learning trained with and without false negatives is more significant on the datasets with more semantic classes.

**Semantic-aware contrastive learning.** To address the limitation of instance-level contrastive learning, several methods are developed to exploit the semantic relationship among images. Chuang et al. (2020) show that the original instance-level sampling strategy includes a sampling bias and propose a debiased contrastive loss. However, this method does not explicitly handle false negative samples. Huynh et al. (2020) determine false negatives of an anchor by finding top $k$ similar samples without referring to global embedding distribution. Khosla et al. (2020) propose a supervised contrastive loss that utilizes label information and treats images of the same class (i.e., false negatives) as positives. However, manually annotated labels cannot be acquired in the unsupervised setting. Closely related to our method, PCL (Li et al., 2021) discovers the representative prototypes for each semantic concept and encourages an image embedding to be closer to its belonging prototype. There are two fundamental differences between PCL and the proposed method: a) PCL uses all assigned prototypes regardless of inferior clustering results in early stages, which cannot properly reflect the semantic relationship among images, b) without treating semantically similar samples as false negatives, PCL applies instance-level InfoNCE loss (Oord et al., 2018) as the primary supervision, which fails to avoid the images within the same prototype being pushed apart. In contrast, leveraging the progressive detection and explicit removal of false negatives, our framework achieves greater clustering quality and representation learning. More comparisons are in Section 4.2 and Appendix E.

**Clustering for deep unsupervised learning.** A few unsupervised learning methods have recently benefited from clustering techniques to construct more effective representation models (Caron et al., 2018; Yang et al., 2016; Xie et al., 2016; Caron et al., 2019; Zhuang et al., 2019; Asano et al., 2020). The main idea is to use cluster indices as pseudo labels and learn visual representations in a supervised manner. However, most existing methods require all pseudo labels to optimize the supervised loss, e.g., cross-entropy. In this work, we show that the pseudo labels obtained in earlier training processes are relatively unreliable, and the full adoption of pseudo labels in the early stage would further disrupt the representation learning. To handle this issue, we incorporate instance-level learning with clustering which allows us to only use a set of pseudo labels with sufficient confidence while treating the others as individual instances.

## 3 METHODOLOGY

### 3.1 INSTANCE-LEVEL CONTRASTIVE LEARNING

An instance-level contrastive learning method learns a representation that discriminates one sample from every other. Given $M$ randomly sampled images from a training set $\mathcal{X}$, a contrastive training mini-batch $\mathcal{I}$ consists of $2M$ images obtained by applying two sets of data augmentation on each sampled image. For any anchor image $i \in \mathcal{I}$, the only positive sample is another view (or transformation) of the same image, denoted as $i'$, while the other $2(M-1)$ images jointly constitute the set of negative samples $\mathcal{N}(i)$. The instance-level discrimination is then optimized by the following contrastive loss:

$$\mathcal{L}_{inst} = \sum_{i \in \mathcal{I}} -\log \frac{\text{sim}(\boldsymbol{z}_i, \boldsymbol{z}_{i'})}{\sum\limits_{s \in \mathcal{S}(i)} \text{sim}(\boldsymbol{z}_i, \boldsymbol{z}_s)}, \quad \mathcal{S}(i) \equiv \{i', n \mid n \in \mathcal{N}(i)\}, \tag{1}$$

where $\boldsymbol{z}_i = g(f(i))$ is the embedding of the image $i$ from an encoder $f$ and a projection head $g$, $\text{sim}(\boldsymbol{u}, \boldsymbol{v}) = \exp(\frac{1}{\tau} \frac{(\boldsymbol{u} \cdot \boldsymbol{v})}{\|\boldsymbol{u}\|\|\boldsymbol{v}\|})$ is the similarity of two input vectors, and $\tau$ represents a temperature hyper-parameter. For instance-level contrastive learning, the negative sample set $\mathcal{N}(i)$ inevitably includes some samples with similar semantic content as the anchor $i$, which we term as false negatives, especially when a larger $M$ is commonly used. Consequently, it would separate semantically similar image pairs, which is suboptimal for learning good semantic-aware visual representations.

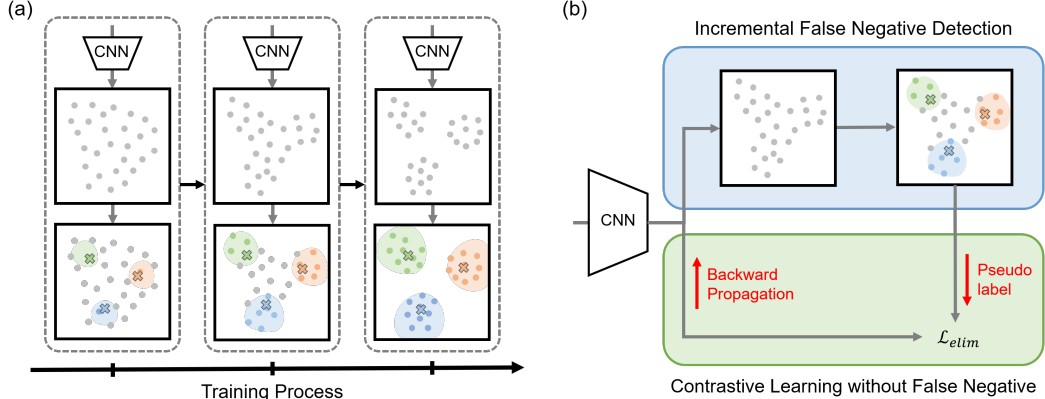

Figure 2: **Algorithmic overview.** (a) Our method uses pseudo labels in an incremental manner with respect to the gradually better-trained encoder and embedding space. (b) The proposed contrastive learning framework uses pseudo labels to explicitly detect and eliminate false negatives from self-supervised contrastive learning.

## 3.2 INCREMENTAL FALSE NEGATIVE DETECTION

To learn an effective semantic-aware representation by self-supervised contrastive learning, our method explicitly detects and removes the false negative samples. Specifically, we follow the same procedure in PCL (Li et al., 2021) to perform $k$-means clustering on the features $f(i)$ of all training images $\{i \in \mathcal{X}\}$ and cluster the features into $k$ groups. We use the centroids to represent the embeddings of the discovered semantic concepts and denote the centroid representation of $k$-th cluster as $c_k$. The pseudo label $y_i$ of the image $i$ is assigned based on the closest centroid, formally $y_i = \arg\min_k \|f(i) - c_k\|$. In the learning process, two images with an identical assigned pseudo label would be treated as a false negative pair.

Nonetheless, as illustrated in the leftmost of Figure 2(a) (or Figure 3 for a realistic case), we observe that the semantic structure of early embedding space is less reliable and could generate noisy pseudo labels, which would disrupt the representation learning. We provide a quantitative analysis of this effect in Section 4.4 and Appendix F. To tackle this problem, in early training stages, we only adopt the pseudo labels with higher confidence while treating the other images as individual samples. For an arbitrary image $i$, the desired confidence $\kappa_i$ should be larger when the image embedding $f(i)$ is not only closer to the centroid of the belonging group, i.e., $c_{yi}$, but also farther to the other centroids. To this end, we measure the confidence of an assigned pseudo label by:

$$\kappa_i = \frac{\text{sim}(\boldsymbol{z}_i, \boldsymbol{c}_{yi})}{\sum_{j=1}^{k} \text{sim}(\boldsymbol{z}_i, \boldsymbol{c}_j)}, \tag{2}$$

which is the softmax of cosine similarities between the image representation and every centroid. We then only assign a specific rate (termed as acceptance rate) of pseudo labels with larger confidence. As plotted in the rest part of Figure 2 (or Figure 3), following the training process, the encoder is better trained and the semantic structure of embedding space is more clear which can generate more reliable pseudo labels. Hence, we propose to incrementally increase the acceptance rate and leverage semantic-aware representation learning. In Section 4.4, we demonstrate that a simple strategy that linearly increases the acceptance rate from $0\%$ to $100\%$ throughout the whole training process can achieve a significant improvement.

**Hierarchical semantic definition.** In an unsupervised setting, the precise class number of a dataset is usually unknown, and also, a different number of classes can be selected by choosing a different level of granularity. To enhance the flexibility and robustness of our method, we incorporate hierarchical clustering techniques (Johnson, 1967; Corpet, 1988; Murtagh & Legendre, 2014; Li et al., 2021) into our framework. Specifically, we perform multiple times of $k$-means with different cluster numbers to define the false negatives in the different levels of granularity. As such, the objective function (will be introduced in Section 3.3) is revised as the average of the original ones computed by the false negatives defined in different levels of granularity.

## 3.3 Contrastive Learning without False Negative

Given the assigned pseudo labels, our next goal is to revise the objective of instance-level contrastive learning, i.e., $\mathcal{L}_{inst}$ in Equation 1, to explicitly deal with the detected false negatives. We discuss two strategies to exploit false negatives. First, *false negative elimination* approach excludes the false negatives from the negative sample set $\mathcal{N}(i)$:

$$\mathcal{L}_{elim} = \sum_{i \in \mathcal{I}} - \log \frac{\text{sim}(\boldsymbol{z}_i, \boldsymbol{z}_{i'})}{\sum\limits_{s \in \mathcal{S}(i)} \text{sim}(\boldsymbol{z}_i, \boldsymbol{z}_s)}, \quad \mathcal{S}(i) \equiv \{i', n \mid n \in \mathcal{N}(i), y_n \neq y_i\}. \tag{3}$$

Second, considering that the anchor and its false negatives are assigned with the same pseudo label, we view the false negatives as additional positive samples:

$$\mathcal{L}_{attr} = \sum_{i \in \mathcal{I}} \frac{1}{|\mathcal{P}(i)|} \sum_{p \in \mathcal{P}(i)} - \log \frac{\text{sim}(\boldsymbol{z}_i, \boldsymbol{z}_p)}{\sum\limits_{s \in \mathcal{S}(i)} \text{sim}(\boldsymbol{z}_i, \boldsymbol{z}_s)}, \begin{cases} \mathcal{S}(i) \equiv \{i', n \mid n \in \mathcal{N}(i)\} \\ \mathcal{P}(i) \equiv \{i', n \mid n \in \mathcal{N}(i), y_n = y_i\}. \end{cases} \tag{4}$$

**Behaviors of elimination and attraction strategies.** Compared to the original contrastive learning objective $\mathcal{L}_{inst}$, $\mathcal{L}_{elim}$ removes the similarities between the anchor and its false negatives from the denominator while on the other hand, $\mathcal{L}_{attr}$ places them on the numerator. They both show a great capability to enlarge the similarities between the anchor and the false negatives when the objective is minimized. To further explore and compare their behaviors, we derive the gradients with regard to the embedding of the anchor $\boldsymbol{z}_i$ as:

$$\frac{\partial \mathcal{L}_{elim,i}}{\partial \boldsymbol{z}_i} = \sum_{\substack{n \in \mathcal{N}(i), \\ y_n \neq y_i}} \frac{\sigma_n^-}{\tau} \boldsymbol{z}_n - \frac{\sigma_{i'}^+}{\tau} \boldsymbol{z}_{i'} \quad \text{and} \quad \frac{\partial \mathcal{L}_{attr,i}}{\partial \boldsymbol{z}_i} = \sum_{\substack{n \in \mathcal{N}(i), \\ y_n \neq y_i}} \frac{\sigma_n^-}{\tau} \boldsymbol{z}_n - \sum_{p \in \mathcal{P}(i)} \frac{\sigma_p^+}{\tau} \boldsymbol{z}_p \tag{5}$$

where $\sigma$ is the weighted coefficient. The detailed formulation and derivation are in Appendix A. Based on their gradients in Equation 5, $\mathcal{L}_{elim}$ and $\mathcal{L}_{attr}$ are the generalization of triplet loss[†] (Schroff et al., 2015) which would intrinsically repel the negatives and attract the positive(s) to the anchor. While both of them fully eliminate the detected false negatives (i.e., $\{n \mid n \in \mathcal{N}(i), y_n = y_i\}$) from the negative samples and avoid them being pushed apart from the anchor, $\mathcal{L}_{attr}$ further minimizes the distances between the anchor and its false negatives. Such a more aggressive strategy results in less tolerance to noisy pseudo labels which are inevitably generated from an unsupervised clustering. Hence, although SupCon (Khosla et al., 2020) presents the effectiveness of $\mathcal{L}_{attr}$ with manual label annotations, we find $\mathcal{L}_{elim}$ is more stable and suitable to our self-supervised framework.

Finally, we illustrate the framework in Figure 2(b) and present the pseudo code in Appendix B.

## 4 Experiments

We validate and analyze our framework on several benchmarks. We train the model on 8 Nvidia V100 GPUs for all experiments. For fair comparisons, we group prior self-supervised algorithms with a similar setting to demonstrate the efficacy of our proposed method.

### 4.1 Implementation Details

We use ResNet-50 (He et al., 2016) as the encoder, followed by a 3-layer MLP as the projection head to obtain 128-D features $\boldsymbol{z}$. We apply the proposed method to ImageNet (Deng et al., 2009) and CIFAR (Krizhevsky et al., 2009). For ImageNet pre-training, we follow the same setting in MoCo v2 (Chen et al., 2020d) and PCL (Li et al., 2021) for fair comparisons. Specifically, we train the encoder for 200 epochs with a mini-batch size of 256 and use the memory-based method (He et al., 2020) to increase the number of contrastive samples in default. For pseudo label assignment, we employ $k$-means clustering implemented by Johnson et al. (2019) to re-assign the pseudo labels after every training epoch and use three cluster numbers $\{1000, 3000, 10000\}$ to define semantic concepts. In default, we linearly increase the pseudo label acceptance rate from $0\%$ to $100\%$ through the whole training process and use the elimination loss $\mathcal{L}_{elim}$ as the objective. The training lasts for 60 hours of which 3.5 hours is spent on pseudo label assignment. More details and the setting for CIFAR pre-training are provided in Appendix C.

---

[†]The gradient of triplet loss $\left( \left[ \|\boldsymbol{z}_i - \boldsymbol{z}_p\|_2^2 - \|\boldsymbol{z}_i - \boldsymbol{z}_n\|_2^2 + \alpha \right]^+ \right)$ w.r.t $\boldsymbol{z}_i$ is $(2\boldsymbol{z}_n - 2\boldsymbol{z}_p)$.

Table 1: **Linear evaluation and transfer learning on three benchmarks.** We report the Top-1 classification accuracy (%). The upper group uses a more compact backbone (AlexNet (Krizhevsky et al., 2012) or ResNet-50 (He et al., 2016)), and smaller pre-training batchsize ($\leqslant 256$).

| Method | Architecture | Pre-training | | Datasets | | |
|---|---|---|---|---|---|---|
| | | batchsize | epochs | ImageNet | VOC | Places |
| Jigsaw (Noroozi & Favaro, 2016) | AlexNet | 256 | - | 34.6 | 67.6 | - |
| Rotation (Gidaris et al., 2018) | AlexNet | 128 | 100 | 38.7 | 73.0 | 35.1 |
| DeepCluster (Caron et al., 2018) | AlexNet | 256 | 500 | 41.0 | 73.7 | 39.8 |
| InstDisc (Wu et al., 2018) | ResNet-50 | 256 | 200 | 54.0 | - | 45.5 |
| LocalAgg (Zhuang et al., 2019) | ResNet-50 | 128 | 200 | 60.2 | - | 50.1 |
| CMC (Tian et al., 2019) | ResNet-50 | - | 200 | 66.2 | - | - |
| SimCLR (Chen et al., 2020b) | ResNet-50 | 256 | 200 | 64.3 | - | - |
| MoCo (He et al., 2020) | ResNet-50 | 256 | 200 | 60.6 | 79.2 | 48.9 |
| MoCo v2 (Chen et al., 2020d) | ResNet-50 | 256 | 200 | 67.5 | 84.0 | 50.1 |
| PCL (Li et al., 2021) | ResNet-50 | 256 | 200 | 67.6 | 85.4 | 50.3 |
| IFND (Ours) | ResNet-50 | 256 | 200 | **69.7** | **87.3** | **51.9** |
| CPC (Oord et al., 2018) | ResNet-101 | 512 | - | 48.7 | - | - |
| SeLa (Asano et al., 2020) | ResNet-50 | 1024 | 400 | 61.5 | - | - |
| PIRL (Misra & Maaten, 2020) | ResNet-50 | 1024 | 800 | 63.6 | 81.8 | 49.8 |
| SimCLR (Chen et al., 2020b) | ResNet-50 | 4096 | 1000 | 69.3 | - | - |
| BYOL (Grill et al., 2020) | ResNet-50 | 4096 | 1000 | 74.3 | - | - |
| SwAV (Caron et al., 2020) | ResNet-50 | 4096 | 800 | 75.3 | 88.9 | 56.7 |

Table 2: **Semi-supervised learning on ImageNet.** We report the Top-5 classification accuracy (%). The numbers of MoCo and MoCo v2 are computed from the official pretrained models.

| Method | Architecture | Pre-training | | Label fraction | |
|---|---|---|---|---|---|
| | | batchsize | epochs | 1% | 10% |
| InstDisc (Wu et al., 2018) | ResNet-50 | 256 | 120 | 39.2 | 77.4 |
| MoCo (He et al., 2020) | ResNet-50 | 256 | 200 | 56.9 | 83.0 |
| MoCo v2 (Chen et al., 2020d) | ResNet-50 | 256 | 200 | 66.3 | 84.4 |
| PCL (Li et al., 2021) | ResNet-50 | 256 | 200 | 75.3 | 85.6 |
| IFND (Ours) | ResNet-50 | 256 | 200 | **77.0** | **86.5** |
| S4L(MOAM) (Zhai et al., 2019) | ResNet-50 ($4\times$) | 256 | 1000 | - | 91.2 |
| PIRL (Misra & Maaten, 2020) | ResNet-50 | 1024 | 800 | 57.2 | 83.8 |
| SimCLR (Chen et al., 2020b) | ResNet-50 | 4096 | 1000 | 75.5 | 87.8 |
| BYOL (Grill et al., 2020) | ResNet-50 | 4096 | 1000 | 78.4 | 89.0 |
| SwAV (Caron et al., 2020) | ResNet-50 | 4096 | 800 | 78.5 | 89.9 |

## 4.2 COMPARISON WITH THE STATE-OF-THE-ARTS

**Linear evaluation and transfer learning on image classification benchmarks.** We first assess the performance of the image representations after pre-training on ImageNet. We train a linear classifier with the fixed representations as input on three benchmarks: ImageNet (Deng et al., 2009), VOC2007 (Everingham et al., 2010) and Places205 (Zhou et al., 2014) and report the results in Table 1. IFND obtains consistent improvements compared to previous self-supervised contrastive learning approaches with a similar experiment setting. The direct comparison to PCL (Li et al., 2021) emphasizes the advantages of incrementally detecting and explicitly removing the false negatives in contrastive learning. We provide a more in-depth ablation study with PCL in Appendix E.

**Semi-supervised learning on ImageNet.** Next, we evaluate the embeddings pre-training on ImageNet in the semi-supervised setting. We follow the protocol described in SimCLR (Chen et al., 2020b) and use the same 1% or 10% fraction of ImageNet labeled training set to finetune the pre-trained encoder. The results are shown in Table 2. IFND outperforms the prior approaches within a similar setup; notably, when using the 1% label, it can achieve more superior performance than SimCLR with 16 times larger batchsize and 5 times more training epochs.

**Object detection and instance segmentation on COCO.** We evaluate our representation on the tasks of object detection and semantic segmentation. Following the experiment setting in MoCo (He et al., 2020) and PCL, we finetune all layers end-to-end on the COCO (Lin et al., 2014) train2017

Table 3: **Object detection and instance segmentation on COCO.** We report bounding-box AP ($AP^{bb}$) and mask AP ($AP^{mk}$) on val2017 (Lin et al., 2014) (%). We use Mask R-CNN (He et al., 2017) with C4 backbone as the model, and the schedule is the default $2\times$ in Girshick et al. (2018). *Supervise* represents the pre-trained model through the supervised training scheme.

| Method | $AP^{bb}$ | $AP^{bb}_{50}$ | $AP^{bb}_{75}$ | $AP^{mk}$ | $AP^{mk}_{50}$ | $AP^{mk}_{75}$ |
|---|---|---|---|---|---|---|
| Supervise | 40.0 | 59.9 | 43.1 | 34.7 | 56.5 | 36.9 |
| MoCo (He et al., 2020) | 40.7 | 60.5 | 44.1 | 35.4 | 57.3 | 37.6 |
| PCL (Li et al., 2021) | 41.0 | 60.8 | 44.2 | 35.6 | 57.4 | 37.8 |
| IFND (Ours) | **41.8** | **61.2** | **44.5** | **36.1** | **57.6** | **38.5** |

Table 4: **Clustering quality on ImageNet.** We compare $k$-means clustering ($k = 1000$) performance for different algorithms (mostly for deep clustering-based methods) using Normalized Mutual Information (Strehl & Ghosh, 2002) (%) for evaluation. We run $k$-means five times and report the average score with the standard deviation. All results are from the official pre-trained models.

| Method | NMI |
|---|---|
| DeepCluster (Caron et al., 2018) | $43.2 \pm 2.9$ |
| MoCo v2 (Chen et al., 2020d) | $57.9 \pm 2.2$ |
| SwAV (Caron et al., 2020) | $63.8 \pm 1.6$ |
| PCL (Li et al., 2021) | $65.0 \pm 1.9$ |
| IFND (Ours) | $\mathbf{67.5} \pm 1.7$ |

Table 5: **Effect of false negatives on self-supervised learning.** We report the Top-1 classification accuracy (%) and use the results of SupCon as the oracle performance with no effect of false negative. $\Delta$ represents the performance drop due to training with false negatives.

| Method | Class label | CIFAR 10 | CIFAR 100 | ImageNet | | | | |
|---|---|---|---|---|---|---|---|---|
| | | | | Depth 3 | Depth 5 | Depth 7 | Depth 9 | Depth 18 |
| SimCLR | ✗ | 94.2 | 71.5 | 93.9 | 86.0 | 75.3 | 71.6 | 65.1 |
| IFND (Ours) | ✗ | 95.1 | 74.0 | 95.2 | 87.5 | 77.2 | 73.8 | 68.4 |
| SupCon | ✓ | 95.9 | 76.2 | 96.1 | 89.2 | 79.3 | 76.3 | 71.5 |
| Number of categories | | 10 | 100 | 9 | 36 | 206 | 486 | 1000 |
| $\Delta$(SupCon, SimCLR) | | -1.7 | -4.7 | -2.2 | -3.2 | -4.0 | -4.7 | -6.4 |
| $\Delta$(SupCon, IFND) | | **-0.8** | **-2.2** | **-0.9** | **-1.7** | **-2.1** | **-2.5** | **-3.1** |

and evaluate on val2017 dataset. We report the results in Table 3. IFND outperforms MoCo, PCL, and supervised pre-training in all metrics.

**Clustering quality on ImageNet.** We also measure the clustering quality on the ImageNet pre-training representations and report the results in Table 4. Compared to MoCo v2, our method leverages semantic-aware learning and shows a significant improvement of $9.6\%$ on NMI. Moreover, IFND achieves the best clustering quality compared to existing deep clustering-based contrastive learning frameworks. In Appendix F, we analyze the clusterings during the training process to better study our incremental manner to apply the clustering assignments.

### 4.3 EFFECT OF FALSE NEGATIVE SAMPLES

**Experimental Setup.** To study the effect of false negatives on self-supervised contrastive learning, we compare three representative frameworks, including SimCLR (Chen et al., 2020b) as instance-level learning, SupCon (Khosla et al., 2020) as supervised learning using label information to completely exclude the false negatives, and our method. For a fair comparison, we reproduce SimCLR and SupCon by using the same data augmentations and network architecture (i.e., ResNet-50 with 3-layer MLP). We also apply SimCLR-based implementation for our method without maintaining a momentum feature bank in this section. Considering that the probability of sampling false negatives in a training mini-batch $\mathcal{I}$ is largely determined by the number of semantic classes, we conduct the experiments on CIFAR (Krizhevsky et al., 2009) and ImageNet (Deng et al., 2009) with different numbers of object categories. For ImageNet, we use WordNet (Miller, 1998) lexical hierarchy to redefine the semantic classes in different depths, as detailed in Appendix D. We then train and perform the linear evaluation for three frameworks on each dataset and report the results in Table 5.

Table 6: **Different strategies for pseudo label assignment and false negative removal on CIFAR-100.** We report MTPR (%, ↑) and MTNR (%, ↑) of $k$-means clustering ($k = 100$) and the Top-1 classification accuracy (%). Note that, [a] is the instance-level learning; [b] and [c] uses the strategy in DeepCluster (Caron et al., 2018) and PCL (Li et al., 2021) (*Step* scheme starts to use the pseudo labels after $100^{th}$ epoch over 1000 training epochs); [g] is the final setting of the proposed method.

| Index | Acceptance Rate of Pseudo Label | | | Objective | MTPR | MTNR | Top-1 Acc. |
|-------|---------|---------|-------|-----------|------|------|------------|
|       | Scheme | Initial | Final | | | | |
| [a] | Constant | 0% | 0% | $\mathcal{L}_{inst}$ | 0 | 100 | 71.5 |
| [b] | Constant | 100% | 100% | $\mathcal{L}_{elim}$ | 31.7 | 99.09 | 71.3 |
| [c] | Step | 0% | 100% | $\mathcal{L}_{elim}$ | 40.0 | 99.42 | 73.1 |
| [d] | Linear | 0% | 25% | $\mathcal{L}_{elim}$ | 15.3 | 99.98 | 72.5 |
| [e] | Linear | 0% | 50% | $\mathcal{L}_{elim}$ | 26.2 | 99.89 | 73.0 |
| [f] | Linear | 0% | 75% | $\mathcal{L}_{elim}$ | 36.0 | 99.78 | 73.5 |
| [g] | Linear | 0% | 100% | $\mathcal{L}_{elim}$ | **43.3** | 99.65 | **74.0** |
| [h] | Linear | 0% | 100% | $\mathcal{L}_{attr}$ | 28.6 | 98.91 | 70.2 |

Based on the performance drops of SimCLR compared to SupCon ($\Delta$(SupCon, SimCLR) in Table 5), we observe that the adverse effect of false negatives is more significant on the datasets with more object categories, although they are less likely to sample false negatives during contrastive learning. Similar results are observed on both CIFAR and ImageNet. The results convey an issue of vanilla instance-level contrastive learning when it is applied to datasets with more complex semantic concepts. With the explicit elimination of the detected false negatives, IFND effectively narrows the gap between self-supervised and supervised frameworks, numerically reducing $51.7\%$ of the gap between SimCLR and SupCon on ImageNet with original labels (i.e., Depth 18).

## 4.4 STRATEGY FOR FALSE NEGATIVE DETECTION AND REMOVAL

In this section, we first introduce a comprehensible evaluation metric for clustering assignment and then present the quantitative study on different strategies to utilize the clustering labels or to remove the detected false negatives. The results on CIFAR-100 are shown in Table 6.

**More interpretable metric for clustering performance.** Considering that the pseudo labels are used to detect false negatives, we propose Mean True Positive/Negative Rate (MTPR/MTNR) to evaluate the clustering. MTPR is the average probability that an actual false negative (i.e., an actual positive) will be detected as a false negative sample, while MTNR indicates the average probability that an actual negative is still treated as a negative sample. Taking the instance-level contrastive learning as a toy example, it does not perform false negative detection, so MTPR is $0\%$ while MTNR is $100\%$. The goal is to increase MTPR of the clustering but keep MTNR as high as possible.

**Strategies for pseudo label assignment.** First, we compare three schemes to determine the acceptance rate of the pseudo labels, including constant ([b], as in DeepCluster (Caron et al., 2018)), step ([c], as in PCL (Li et al., 2021)), and linear ([g], *Ours*) schemes. The linear strategy achieves both higher MTPR and MTNR. The results validate that using the clustering indices in an incremental manner can find the false negatives more effectively and further improve the performance of contrastive learning. Next, we analyze the effect of using different final acceptance rates for the linear strategy ([d,e,f,g]). Intuitively, higher acceptance rates indicate more false negatives can be detected (MTPR increases), but more true negatives are also improperly treated as false negative samples and removed (MTNR decreases). However, we find that using the $100\%$ final acceptance rate ([g]) can averagely detect nearly a half ($43.3\%$) of the actual false negatives with only a slight drop ($0.35\%$) of MTNR compared to the instance-level approach ([a]). Along with the elimination loss, we improve the downstream Top-1 classification accuracy from $71.5\%$ to $74.0\%$.

**Strategies for false negative removal.** Finally, we compare the elimination and attraction losses ([g,h]). We find that the representation learning through minimizing $\mathcal{L}_{attr}$ are unstable and rely on the high-quality pseudo labels at the early training stages. This observation can be explained by the theoretical analysis in Section 3.3 indicating that $\mathcal{L}_{attr}$ applies a more aggressive strategy that would be more sensitive to noisy pseudo labels. Empirically, we observe that the model supervised by $\mathcal{L}_{attr}$ performs even worse than the instance-level learning.

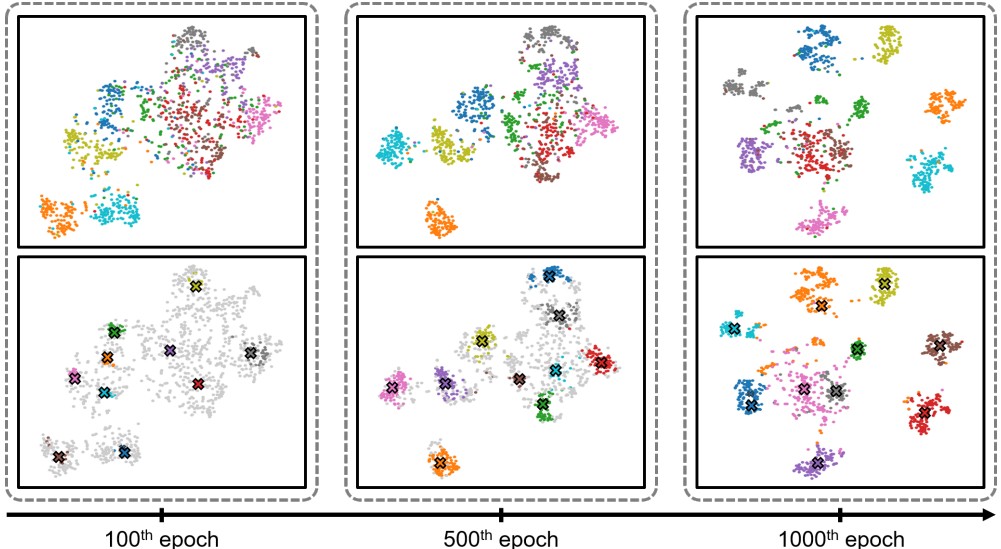

Figure 3: **Visualization of embedding spaces and pseudo label assignments in different training stages on CIFAR-10.** The samples in the upper and lower row are respectively colored with the ground truth labels and the assigned pseudo labels. ✖ marks are the cluster centroids, and light grey dots represent the instances without the assigned pseudo labels.

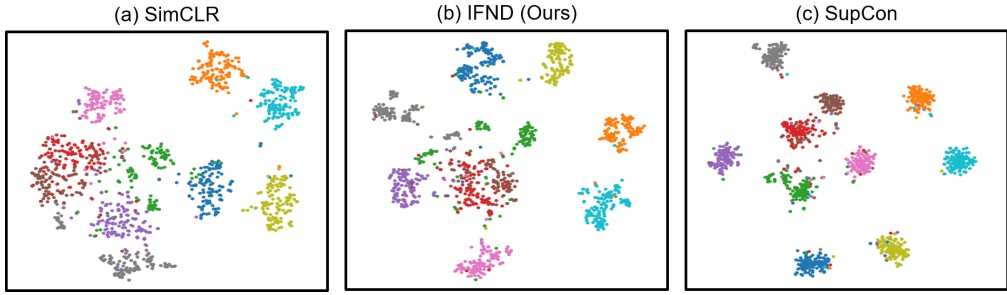

Figure 4: **Visualization of the embedding spaces learned by different methods on CIFAR-10.**

## 4.5 VISUALIZATION OF EMBEDDING SPACE

In Figure 3, we visualize the embedding spaces and corresponding pseudo label assignments in different training stages using t-SNE (Van der Maaten & Hinton, 2008). In the early stages (e.g., $100^{th}$ epoch), the semantic structure of the embedding space is still under-developed. Hence, IFND only adopts a small portion of pseudo labels with higher confidence scores. Later in the training process, with the better-trained encoder and more semantically structural embedding space, incrementally including more high-quality pseudo labels can benefit the semantic-aware representation learning. Finally, we compare the embedding spaces learned through SimCLR, SupCon, and our approach in Figure 4. IFND can learn the embedding space with more compact clusters than the instance-level learning and is also more similar to the supervised framework. We present a quantitative analysis in Appendix F and also provide more visualizations and discussions in Appendix G.

## 5 CONCLUSION

In this work, we analyze the undesired effects of false negatives for self-supervised contrastive learning. To mitigate such adverse effects, we then introduce an incremental false negative detection approach to progressively detect and remove the false negatives through the training stage. Extensive experiment results validate the effectiveness of the proposed framework on several benchmarks and also demonstrate the capability to reduce the gap from the supervised framework.

ACKNOWLEDGEMENTS

This work is supported in part by the NSF CAREER grant 1149783.

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

## A   THEORETICAL ANALYSIS FOR ELIMINATION AND ATTRACTION LOSSES

In this section, we show the derivation of the gradients of two discussed objective functions: elimination loss $\mathcal{L}_{elim}$ (Equation 3) and attraction loss $\mathcal{L}_{attr}$ (Equation 4). Furthermore, we demonstrate that they have the inherent capability of hard sample mining (Oh Song et al., 2016; Harwood et al., 2017; Wu et al., 2017). Without loss of generality, we use $i$ to represent an arbitrary anchor image, and re-write the two losses as:

$$\mathcal{L}_{elim,i} = -\log \frac{\exp(\boldsymbol{z}_i \cdot \boldsymbol{z}_{i'}/\tau)}{\sum\limits_{s \in \mathcal{S}(i)} \exp(\boldsymbol{z}_i \cdot \boldsymbol{z}_s/\tau)}, \quad \mathcal{S}(i) \equiv \{i', n \mid n \in \mathcal{N}(i), y_n \neq y_i\}, \quad \text{and} \quad (6)$$

$$\mathcal{L}_{attr,i} = \frac{1}{|\mathcal{P}(i)|} \sum_{p \in \mathcal{P}(i)} -\log \frac{\exp(\boldsymbol{z}_i \cdot \boldsymbol{z}_p/\tau)}{\sum\limits_{s \in \mathcal{S}(i)} \exp(\boldsymbol{z}_i \cdot \boldsymbol{z}_s/\tau)}, \begin{cases} \mathcal{S}(i) \equiv \{i', n \mid n \in \mathcal{N}(i)\} \\ \mathcal{P}(i) \equiv \{i', n \mid n \in \mathcal{N}(i), y_n = y_i\}. \end{cases}$$

$$(7)$$

We then derive the gradient of $\mathcal{L}_{elim,i}$ with regard to the embedding of the anchor image $\boldsymbol{z}_i$:

$$\begin{aligned}
\frac{\partial \mathcal{L}_{elim,i}}{\partial \boldsymbol{z}_i} &= \frac{\partial}{\partial \boldsymbol{z}_i} -\log \frac{\exp(\boldsymbol{z}_i \cdot \boldsymbol{z}_{i'}/\tau)}{\sum\limits_{s \in \mathcal{S}(i)} \exp(\boldsymbol{z}_i \cdot \boldsymbol{z}_s/\tau)} \\
&= \frac{\partial}{\partial \boldsymbol{z}_i} \log \sum_{s \in \mathcal{S}(i)} \exp(\boldsymbol{z}_i \cdot \boldsymbol{z}_s/\tau) - \frac{\partial}{\partial \boldsymbol{z}_i} \log(\exp(\boldsymbol{z}_i \cdot \boldsymbol{z}_{i'}/\tau)) \\
&= \sum_{s \in \mathcal{S}(i)} \left( \frac{\exp(\boldsymbol{z}_i \cdot \boldsymbol{z}_s/\tau)}{\sum\limits_{s \in \mathcal{S}(i)} \exp(\boldsymbol{z}_i \cdot \boldsymbol{z}_s/\tau)} \cdot \frac{\boldsymbol{z}_s}{\tau} \right) - \frac{\boldsymbol{z}_{i'}}{\tau} \\
&= \sum_{\substack{n \in \mathcal{N}(i), \\ y_n \neq y_i}} \left( \frac{\exp(\boldsymbol{z}_i \cdot \boldsymbol{z}_n/\tau)}{\sum\limits_{s \in \mathcal{S}(i)} \exp(\boldsymbol{z}_i \cdot \boldsymbol{z}_s/\tau)} \cdot \frac{\boldsymbol{z}_n}{\tau} \right) - \left( 1 - \frac{\exp(\boldsymbol{z}_i \cdot \boldsymbol{z}_{i'}/\tau)}{\sum\limits_{s \in \mathcal{S}(i)} \exp(\boldsymbol{z}_i \cdot \boldsymbol{z}_s/\tau)} \right) \cdot \frac{\boldsymbol{z}_{i'}}{\tau} \\
&= \sum_{\substack{n \in \mathcal{N}(i), \\ y_n \neq y_i}} \frac{\sigma_n^-}{\tau} \boldsymbol{z}_n - \frac{\sigma_{i'}^+}{\tau} \boldsymbol{z}_{i'},
\end{aligned}$$

$$(8)$$

where

$$\begin{aligned}
\sigma_x^- &= \frac{\exp(\boldsymbol{z}_i \cdot \boldsymbol{z}_x/\tau)}{\sum\limits_{s \in \mathcal{S}(i)} \exp(\boldsymbol{z}_i \cdot \boldsymbol{z}_s/\tau)} = \frac{\text{sim}(\boldsymbol{z}_i, \boldsymbol{z}_x)}{\sum\limits_{s \in \mathcal{S}(i)} \text{sim}(\boldsymbol{z}_i \cdot \boldsymbol{z}_s)}, \quad \text{and} \\
\sigma_x^+ &= 1 - \frac{\exp(\boldsymbol{z}_i \cdot \boldsymbol{z}_x/\tau)}{\sum\limits_{s \in \mathcal{S}(i)} \exp(\boldsymbol{z}_i \cdot \boldsymbol{z}_s/\tau)} = 1 - \frac{\text{sim}(\boldsymbol{z}_i, \boldsymbol{z}_x)}{\sum\limits_{s \in \mathcal{S}(i)} \text{sim}(\boldsymbol{z}_i \cdot \boldsymbol{z}_s)}
\end{aligned}$$

$$(9)$$

are the coefficient terms respectively for the negative or positive sample $x$.

Next, the gradient of $\mathcal{L}_{attr,i}$ is derived in a similar way:

$$
\begin{aligned}
\frac{\partial \mathcal{L}_{attr,i}}{\partial \boldsymbol{z}_i} &= \frac{\partial}{\partial \boldsymbol{z}_i} \frac{1}{|\mathcal{P}(i)|} \sum_{p \in \mathcal{P}(i)} -\log \frac{\exp(\boldsymbol{z}_i \cdot \boldsymbol{z}_p / \tau)}{\sum\limits_{s \in \mathcal{S}(i)} \exp(\boldsymbol{z}_i \cdot \boldsymbol{z}_s / \tau)} \\
&= \frac{1}{|\mathcal{P}(i)|} \sum_{p \in \mathcal{P}(i)} \left( \frac{\partial}{\partial \boldsymbol{z}_i} \log \sum_{s \in \mathcal{S}(i)} \exp(\boldsymbol{z}_i \cdot \boldsymbol{z}_s / \tau) - \frac{\partial}{\partial \boldsymbol{z}_i} \log \exp(\boldsymbol{z}_i \cdot \boldsymbol{z}_p / \tau) \right) \\
&= \frac{1}{|\mathcal{P}(i)|} \sum_{p \in \mathcal{P}(i)} \left[ \sum_{s \in \mathcal{S}(i)} \left( \frac{\exp(\boldsymbol{z}_i \cdot \boldsymbol{z}_s / \tau)}{\sum\limits_{s \in \mathcal{S}(i)} \exp(\boldsymbol{z}_i \cdot \boldsymbol{z}_s / \tau)} \cdot \frac{\boldsymbol{z}_s}{\tau} \right) - \frac{\boldsymbol{z}_p}{\tau} \right] \\
&= \sum_{s \in \mathcal{S}(i)} \left( \frac{\exp(\boldsymbol{z}_i \cdot \boldsymbol{z}_s / \tau)}{\sum\limits_{s \in \mathcal{S}(i)} \exp(\boldsymbol{z}_i \cdot \boldsymbol{z}_s / \tau)} \cdot \frac{\boldsymbol{z}_s}{\tau} \right) - \sum_{p \in \mathcal{P}(i)} \frac{1}{|\mathcal{P}(i)|} \frac{\boldsymbol{z}_p}{\tau} \qquad (10) \\
&= \sum_{\substack{n \in \mathcal{N}(i), \\ y_n \neq y_i}} \left( \frac{\exp(\boldsymbol{z}_i \cdot \boldsymbol{z}_n / \tau)}{\sum\limits_{s \in \mathcal{S}(i)} \exp(\boldsymbol{z}_i \cdot \boldsymbol{z}_s / \tau)} \cdot \frac{\boldsymbol{z}_n}{\tau} \right) \\
&\qquad\qquad\qquad\qquad - \sum_{p \in \mathcal{P}(i)} \left( \frac{1}{|\mathcal{P}(i)|} - \frac{\exp(\boldsymbol{z}_i \cdot \boldsymbol{z}_p / \tau)}{\sum\limits_{s \in \mathcal{S}(i)} \exp(\boldsymbol{z}_i \cdot \boldsymbol{z}_s / \tau)} \right) \cdot \frac{\boldsymbol{z}_p}{\tau} \\
&= \sum_{\substack{n \in \mathcal{N}(i), \\ y_n \neq y_i}} \frac{\sigma_n^-}{\tau} \boldsymbol{z}_n - \sum_{p \in \mathcal{P}(i)} \frac{\sigma_p^+}{\tau} \boldsymbol{z}_p,
\end{aligned}
$$

where

$$
\begin{aligned}
\sigma_x^- &= \frac{\exp(\boldsymbol{z}_i \cdot \boldsymbol{z}_x / \tau)}{\sum\limits_{s \in \mathcal{S}(i)} \exp(\boldsymbol{z}_i \cdot \boldsymbol{z}_s / \tau)} = \frac{\mathrm{sim}(\boldsymbol{z}_i, \boldsymbol{z}_x)}{\sum\limits_{s \in \mathcal{S}(i)} \mathrm{sim}(\boldsymbol{z}_i \cdot \boldsymbol{z}_s)}, \quad \text{and} \\
\sigma_x^+ &= \frac{1}{|\mathcal{P}(i)|} - \frac{\exp(\boldsymbol{z}_i \cdot \boldsymbol{z}_x / \tau)}{\sum\limits_{s \in \mathcal{S}(i)} \exp(\boldsymbol{z}_i \cdot \boldsymbol{z}_s / \tau)} = \frac{1}{|\mathcal{P}(i)|} - \frac{\mathrm{sim}(\boldsymbol{z}_i, \boldsymbol{z}_x)}{\sum\limits_{s \in \mathcal{S}(i)} \mathrm{sim}(\boldsymbol{z}_i \cdot \boldsymbol{z}_s)}.
\end{aligned} \qquad (11)
$$

Note that the coefficient terms $\sigma_x^+$ and $\sigma_x^-$ in Equation 9 and 11 are determined by $\mathrm{sim}(\boldsymbol{z}_i, \boldsymbol{z}_x)$. Larger coefficients are given when negative/positive samples are relatively more similar/dissimilar to the anchor. This observation indicates that these two loss functions have the implicit capability of hard sample mining, i.e., focusing on differentiating the "hard negatives" that are near the anchor and attracting the "hard positives" which are far away from the anchor.

# B PSEUDO CODE

---

**Algorithm 1:** Contrastive Learning with Incremental False Negative Detection

---

**Input:** encoder $f$, projection head $g$, training samples $\mathcal{X}$, and number of clusters $k$
           `// instance-level contrastive learning at the beginning`

1   Initialize completely different pseudo labels $y^*$ for each sample
2   **while** *Epoch $\in$ Training Epoch* **do**
              `// contrastive learning with false negative elimination`
3      **for** *mini-batch $\mathcal{I}$* **do**
4         $z = g(f(\{i \sim \mathcal{I}\}))$
5         update encoder $f$ and projection head $g$ to minimize $\mathcal{L}_{elim}(z, y^*)$
6      **end**
7      **if** *Epoch $\in$ Clustering Epoch* **then**           `// pseudo label assignment`
8         $v = f(\{x \sim \mathcal{X}\})$
9         $c, y = k\text{-means}(v, k)$
10        $\kappa = \text{sim}(z, c_y) / \sum_{j=1}^{k} \text{sim}(z, c_j)$
11        determine the acceptance rate of the pseudo label as (*Epoch / Training Epoch*)
12        update pseudo labels $y^*$ by a part of $y$ with larger $\kappa$
13      **end**
14   **end**

---

# C PRE-TRAINING AND EVALUATION DETAILS

As described in Section 4.1, we use the proposed method to pre-train ResNet-50 (He et al., 2016) on ImageNet (Deng et al., 2009), CIFAR-10, and CIFAR-100 (Krizhevsky et al., 2009). For ImageNet pre-training, we follow the setting of previous work (He et al., 2020; Chen et al., 2020d) that uses SGD optimizer (Kiefer et al., 1952) with a weight decay of $0.0001$ and a momentum of $0.9$. The learning rate is set to $0.03$ initially and then multiplied by $0.1$ at 120 and 160 epochs. For CIFAR-10 and CIFAR-100 pre-training, we train the encoder for 1000 epochs with a mini-batch size of 1024. Because of the larger mini-batch size, we implement our framework based on SimCLR (Chen et al., 2020b) structure without maintaining a memory bank for the momentum instance features. We also use SGD optimizer with the same weight decay and momentum. But, we set the initial learning rate to $0.5$ and use a linear warm-up strategy for the first 10 epochs. In the rest of the training, we decay the learning rate with the cosine annealing schedule (Loshchilov & Hutter, 2016). For pseudo label assignment, we re-assign the pseudo labels after every 20 training epochs (due to more training epochs and fewer training images) for CIFAR and also use three cluster numbers, $\{10, 30, 100\}$ for CIFAR-10 and $\{100, 300, 1000\}$ for CIFAR-100, to hierarchically define semantic concepts.

For linear evaluation on ImageNet, we train the classifier (a fully connected layer followed by softmax) with the fixed pre-trained representations for 100 epochs. Following the same hyper-parameter setting used in MoCo v2 (Chen et al., 2020d) and PCL (Li et al., 2021), we optimize the model by SGD optimizer with a batch size of 256, an initial learning rate of 10, a momentum of $0.9$, and a weight decay of 0. As for semi-supervised learning, we fine-tune the pre-trained ResNet-50 model and the classifier using the subsets of $1\%$ or $10\%$ ImageNet labeled training data (note that we use the same subsets of ImageNet training data as in SimCLR). The model is trained using SGD optimizer for 20 epochs, a batch size of 256, a momentum of $0.9$, and a weight decay of $0.0005$. The learning rate for ResNet-50 is $0.01$ while the one for the linear classifier is $0.1$ (for $10\%$ ImageNet) or 1 (for $1\%$ ImageNet). All learning rates are multiplied by $0.2$ at $12^{nd}$ and $16^{th}$ epochs.

# D HIERARCHICAL CLASS DEFINITION FOR IMAGENET

Based on WordNet (Miller, 1998) lexical database, we use the corresponding synset IDs to represent original 1000 classes in ImageNet and recursively trace the "hypernyms" (parent synset) for each class until the coarsest synset, "entity", in WordNet. Then, we construct the tree structure with entity synset as the root, the synsets of original classes as the leaves, and the relations of hypernym or hyponym as the edges. We then re-define the semantic class with different granularities in different

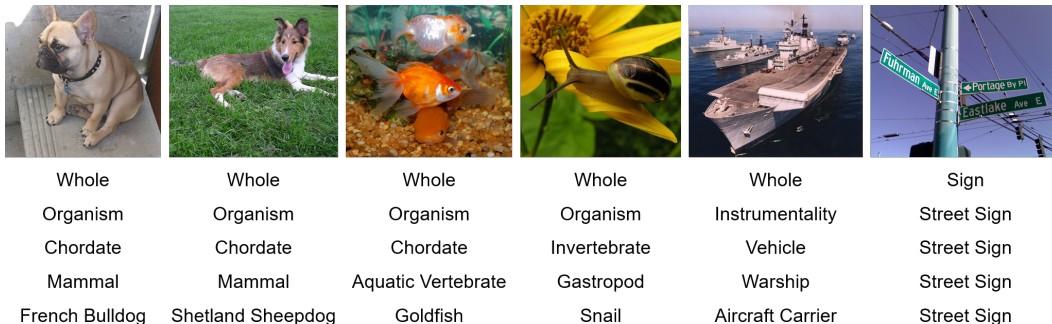

| Whole | Whole | Whole | Whole | Whole | Sign |
| Organism | Organism | Organism | Organism | Instrumentality | Street Sign |
| Chordate | Chordate | Chordate | Invertebrate | Vehicle | Street Sign |
| Mammal | Mammal | Aquatic Vertebrate | Gastropod | Warship | Street Sign |
| French Bulldog | Shetland Sheepdog | Goldfish | Snail | Aircraft Carrier | Street Sign |

Figure 5: **Hierarchical semantic concepts for ImageNet.** We show six images and their respective class labels hierarchically defined in Depth = $\{3, 5, 7, 9, 18\}$ from top to bottom. Depth 18 (the bottommost) uses original class labels in ImageNet.

Table 7: **Ablation comparison with PCL (Li et al., 2021) on CIFAR-100.** [†] indicates the baseline model. We mark the component which is different from the baseline in red and compute the performance gaps to the baseline. The implementation of $\mathcal{L}_{\mathrm{ProtoNCE}}$ and confidence estimation in PCL are directly from the official repository.

| Method | Objective | Pseudo Label Assignment | Numbers of Clusters ($\times 100$) | Confidence Estimation | Top-1 Acc. (%) |
|---|---|---|---|---|---|
| SimCLR | $\mathcal{L}_{inst}$ | Constant | - | - | 71.5 |
| PCL | $\mathcal{L}_{\mathrm{ProtoNCE}}$ | Step | $\{25, 50, 100\}$ | Eq. 9 in PCL | 72.8 |
| **IFND (Ours)** | $\mathcal{L}_{elim}$ | Linear | $\{1, 3, 10\}$ | Eq. 2 | **74.0** |
| PCL[†] | $\mathcal{L}_{elim}$ | Step | $\{25, 50, 100\}$ | Eq. 9 in PCL | 73.1 (0.3↑) |
| IFND[†] | $\mathcal{L}_{\mathrm{ProtoNCE}}$ | Linear | $\{1, 3, 10\}$ | Eq. 2 | 73.4 (0.6↓) |
| PCL[†] | $\mathcal{L}_{\mathrm{ProtoNCE}}$ | Linear | $\{25, 50, 100\}$ | Eq. 9 in PCL | 73.5 (0.7↑) |
| IFND[†] | $\mathcal{L}_{elim}$ | Step | $\{1, 3, 10\}$ | Eq. 2 | 73.1 (0.9↓) |
| PCL[†] | $\mathcal{L}_{\mathrm{ProtoNCE}}$ | Step | $\{1, 3, 10\}$ | Eq. 9 in PCL | 69.9 (2.9↓) |
| IFND[†] | $\mathcal{L}_{elim}$ | Linear | $\{25, 50, 100\}$ | Eq. 2 | 73.2 (0.8↓) |
| IFND[†] | $\mathcal{L}_{elim}$ | Linear | $\{1, 3, 10\}$ | Eq. 9 in PCL | 71.2 (2.8↓) |

depths of the tree. In Section 4.3, we employ the datasets defined in Depth = $\{3, 5, 7, 9, 18\}$. Here, we show some example images with the semantic classes defined in these depths in Figure 5.

## E  IN-DEPTH ABLATION COMPARISON WITH PCL

Here, we provide an ablation comparison with PCL (Li et al., 2021) in terms of 1) objective function, 2) pseudo label assignment, 3) numbers of clusters, and 4) confidence estimation. The experimental setup is the same as in Section 4.3. Briefly, we re-implement PCL and our IFND with SimCLR (Chen et al., 2020b) based implementation and evaluate the frameworks on CIFAR-100 (Krizhevsky et al., 2009). The results are reported in Table 7.

**Objective function.** PCL applies $\mathcal{L}_{\mathrm{ProtoNCE}}$ which includes instance-level infoNCE loss (Gutmann & Hyvärinen, 2010), i.e., $\mathcal{L}_{inst}$, and would suffer from the false negative effect. In contrast, IFND applies $\mathcal{L}_{elim}$ that explicitly eliminates the detected false negatives from negative sample set (as proven in Section 3.3) and performs favorably against $\mathcal{L}_{\mathrm{ProtoNCE}}$.

**Pseudo label assignment.** For the utilization of clustering indices, PCL uses *Step* scheme that first warms up the framework by instance-level learning and adopts all indices afterward. On the other hand, IFND progressively includes increasing confident labels, which better fits the gradually well-trained encoder (as demonstrated in Appendix F).

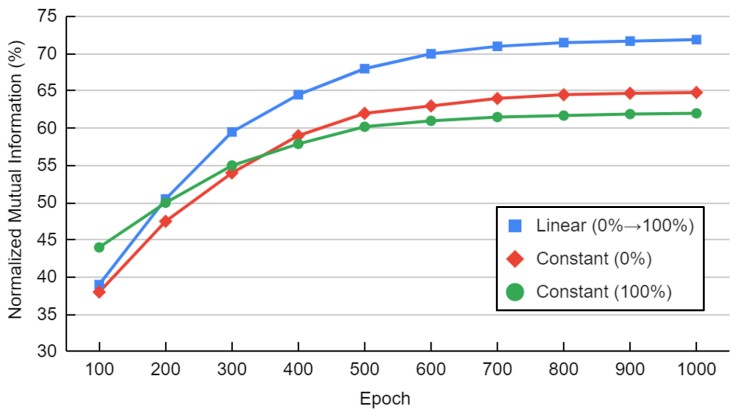

Figure 6: **Clustering quality in different training stages on CIFAR-100.** We train the model with three different strategies to determine the acceptance rate of pseudo labels and measure the quality of $k$-means clusterings ($k = 100$) on the embedding spaces during the training process.

**Numbers of clusters.** The performance of the PCL approach degrades significantly when the number of clusters becomes smaller while the proposed method is more robust to the number of clusters. It is worth mentioning that with smaller numbers of clusters, PCL performs similarly to the attraction strategy (69.9% versus 70.2% as in Table 6).

**Confidence estimation.** PCL estimates a single confidence score for the instances within the same cluster. On the contrary, our method computes the confidence score for each individual instance and also refers to the centroids of other clusters for the estimation. As a result, the representation learning with our confidence estimation performs more favorably.

## F  CLUSTERING QUALITY THROUGH TRAINING PROCESS

In this section, we measure the clustering results in different training stages to first verify the claim that the clusterings are comparably unreliable in the early training stages and also study the practical effect of our incremental manner to adopt clustering indices. We compute Normalized Mutual Information (Strehl & Ghosh, 2002) between the ground truth labels and the clustering indices of all training images and plot the results in Figure 6.

We can observe that following the training, the clustering is gradually improved and becomes better to reflect the semantic relationship among images. Hence, for the model that consistently uses all pseudo labels (green line in Figure 6), the earlier inferior labels would disturb the training. In contrast, the proposed model (blue line in Figure 6) progressively applies increasing high-quality labels that can benefit semantic-aware learning and finally achieve both better representation learning and clustering as in Table 6 and Figure 6.

## G  VISUALIZATION OF EMBEDDING SPACE THROUGH TRAINING PROCESS

In this section, we visualize and compare the embedding spaces learned by four contrastive learning frameworks, including (a) SimCLR (Chen et al., 2020b), (b) IFND (trained by $\mathcal{L}_{elim}$), (c) IFND (trained by $\mathcal{L}_{attr}$), and (d) SupCon (Khosla et al., 2020), through the training process in Figure 7.

Compared to IFND trained by $\mathcal{L}_{elim}$, the one trained by $\mathcal{L}_{attr}$ learns the embedding space with more concentrated clusters, which are more similar to SupCon due to using similar attraction objective function. However, during training, the unsupervised clustering would inevitably produce some noisy pseudo labels (as supported by the MTNR in Table 6 being less than $100\%$). Consequently, the attraction loss may inappropriately pull actual negative samples closer to the anchor. We can then find that although the embedding space in the rightmost of Figure 7(c) shows more concentrated clusters, nonetheless, a cluster is not constituted by the image embeddings of the same semantic

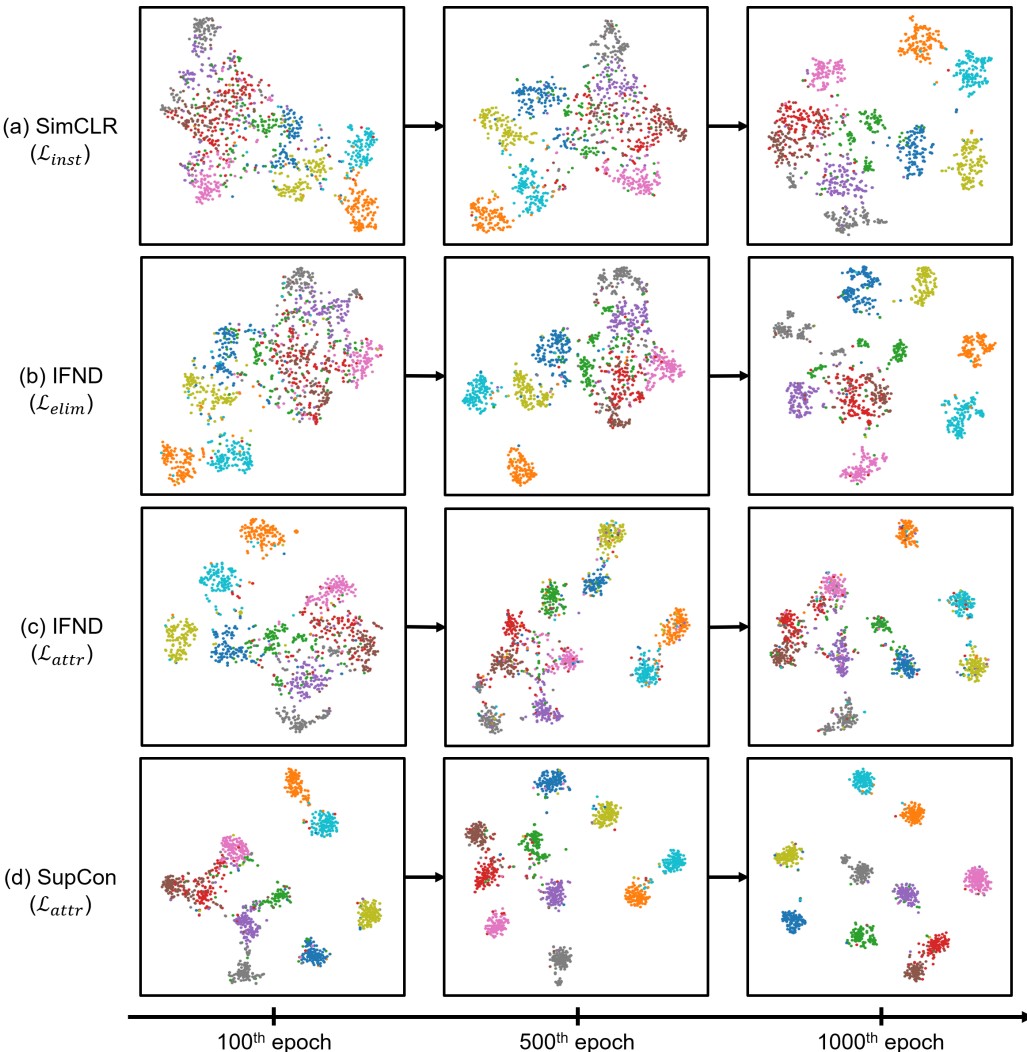

Figure 7: **Visualization of embedding spaces on CIFAR-10.** We compare four contrastive learning frameworks based on their embedding spaces through the training process. We list their objective functions in the bracket.

content. Also, as shown in Table 6, the attraction loss leads to inferior representation performance compared to the elimination objective.

For the final proposed method, i.e., IFND trained by $\mathcal{L}_{elim}$, it learns a similar embedding space as SimCLR in the early stage since the acceptance rate for assigning the pseudo label is still low. Following the training process, with increasing reliable pseudo labels, the proposed framework effectively bridges the instance-level and semantic-aware representation learning and learns a more favorable embedding space against SimCLR.

