# OpenReview forum: "Incremental False Negative Detection for Contrastive Learning"
_ICLR.cc/2022/Conference — ICLR 2022 Poster_

### Official Review · Reviewer_qg7u · 2021-10-24

**Correctness:** 3
**Technical Novelty And Significance:** 3
**Empirical Novelty And Significance:** 2
**Recommendation:** 6
**Confidence:** 4

**Main Review:**

Strengths:
1) Well written and easy to follow paper with a very solid hypothesis and analysis of effects of “false negatives” on contrastive learning.
2) They have shown good results specially under classification settings in both supervised learning and semi-supervised learning setups.
3) Table4 shows really good analysis and shows how increasing the depth results in more and more poor performances.
4) Better clustering results than SWaV: Table 3 results are very promising as they show that IFND achieves better NMI results than SWaV and MoCo-V2. This means that IFND is really able to form better clusters and show improved performances.

Weakness:
1) Results on BYOL and other non-contrastive methods: The method would be much more comprehensive if this method can be extended to other non-contrastive methods such BYOL as well. Recently BYOL and other non-contrastive methods are showing much more promise than contrastive methods. Hence changing the loss function using similar motivation from IFND would be really good to see.
2) Object-Detection and Semantic Segmentation results on COCO/VOC: I couldn't find any results on object detection or segmentation. Adding those results would make the paper more complete.
3) Another interesting result would be to see the performance of IFND by pre-training on the COCO dataset. ImageNet in a way is already clustered and finding close “false negatives” would be easier on ImageNEt as compared to COCO. Results by pre-training on COCO would make the method much more convincing.


**Summary Of The Paper:**

This paper deals with the very important topic of “false negatives” in SSL and shows that “false negatives” have a huge impact on performance of SSL methods. One of the most interesting experiments they show is that as they increase the number of classes in the dataset the performance drops more and more. They propose a clustering based method which reduces the  “false negatives” and shows improved performances as compared to other state-of-the-art baselines.


**Summary Of The Review:**

I like the idea used in the paper and analysis shown in the paper. However lack of results on object detection and semantic segmentation coupled with lack of results using BYOL method, I'm leaning towards borderline rejection. Adding these results would make the paper stronger.

---

> ### Author Response · Authors · 2021-11-22
> **Responses to Reviewer qg7u**
>
> We thank the reviewer for recognizing the good motivation, analysis, and experiment results in our paper. Here are our responses to the reviewer's comments.
>
> ---
>
> > "More comprehensive if this method can be extended to other methods"
>
> BYOL only minimizes the similarity between two representations extracted from the same image, so we cannot directly apply our concept to this approach. As a fallback solution, we extend the proposed idea of incremental false negative removal to the SwAV framework. The idea is to prevent that the images with the same semantic concept are assigned with different codes when SwAV computes the codes online. The experiment results are shown in the below table (the framework in the brackets represents the baseline model).
>
> | Method  | Top-1 Acc. |
> | :-----: | :-----: |
> | `MoCo v2` | 67.5 |
> | `IFND (MoCo v2)` | 69.7 (2.2%$\uparrow$) |
> | `SwAV` | 72.0 |
> | `IFND (SwAV)` | 72.8 (0.8%$\uparrow$) |
>
> By using either MoCo v2 or SwAV as the baseline learning framework, IFND can consistently boost performance. The results demonstrate the advantage of progressively removing the false negative samples. More implementation details and discussions are added in Appendix F of the revised manuscript (revised text is marked in red).
>
> ---
>
> > "Cannot find any results on object detection or segmentation"
>
> As suggested, we finetune the model pre-trained with our method on the COCO train2017 and evaluate on val2017 dataset. The results are shown in the below table and have been added to Section 4.2 in the revised paper.
>
> | Method  | AP$^\mathrm{bb}$ | AP$^\mathrm{bb}_\mathrm{50}$ | AP$^\mathrm{bb}_\mathrm{75}$ | AP$^\mathrm{mk}$ | AP$^\mathrm{mk}_\mathrm{50}$ | AP$^\mathrm{mk}_\mathrm{75}$ |
> | :-----: | :-----: | :-----: | :-----: | :-----: | :-----: | :-----: |
> | `Supervise` | 40.0 | 59.9 | 43.1 | 34.7 | 56.5 | 36.9 |
> | `MoCo` | 40.7 | 60.5 | 44.1 | 35.4 | 57.3 | 37.6 |
> | `PCL` | 41.0 | 60.8 | 44.2 | 35.6 | 57.4 | 37.8 |
> | `IFND (Ours)` | **41.8** | **61.2** | **44.5** | **36.1** | **57.6** | **38.5** |
>
> Overall, the proposed IFND approach performs favorably against the PCL and supervised pre-training schemes on both the object detection and semantic segmentation tasks.
>
> ---
>
> > "Results by pre-training on COCO"
>
> Most of the existing self-supervised learning work (e.g., SimCLR, MoCo, SwAV, BYOL, SimSiam) focuses on learning the image representations on object-centric datasets, such as ImageNet. The representation learning on the complicated scene datasets, where the images contain multiple objects with complex relationships, remains a challenging problem. We leave this problem as one of our future research directions.
>
> [Update in 11/29]
>
> We extend IFND to multi-object datasets using dense contrastive learning [1] as a baseline framework.
>
> In [1], the authors define positive/negative samples for multi-object images in a contrastive learning framework. Specifically, they discard the avgpool layer in ResNet and define the "instance" as the feature extracted from a specific region of the image.
> As such, the false negatives in this work are defined as: the features extracted from other regions which are semantically similar to the anchor region. With this definition, we can extend IFND to multi-object datasets.
>
> We present the experiment for this idea as follows.
> We perform clustering on the embedding space after the dense projection head, where "instance" is defined as a feature extracted from a specific region in the image.
> In order to remove the false negative samples, we revise the instance-level loss (Eq. (2) in DenseCL) to our elimination loss.
> We follow the training details described in their paper but only pre-train the model in 100 epochs due to time constraints.
> The pre-training model is trained on train2017 and evaluated on val2017 dataset. The results are shown in the below table.
>
> | Method  | AP$^\mathrm{bb}$ | AP$^\mathrm{bb}_\mathrm{50}$ | AP$^\mathrm{bb}_\mathrm{75}$ | AP$^\mathrm{mk}$ | AP$^\mathrm{mk}_\mathrm{50}$ | AP$^\mathrm{mk}_\mathrm{75}$ |
> | :-----: | :-----: | :-----: | :-----: | :-----: | :-----: | :-----: |
> | `Dense CL*` | 38.7 | 58.2 | 42.0 | 34.5 | 55.1 | 37.0 |
> | `IFND (Ours)` | **38.9** | **58.6** | **42.3** | **34.6** | **55.5** | **37.6** |
>
> *We reproduce DenseCL by training 100 epochs for a fair comparison.
>
> The results illustrate that the proposed IFND can also be applied to multi-object datasets.
> We will add more details and experiment results in the final paper.
> Please let us know whether you have additional questions or not.
>
> Thank you,
>
> [1] X. Wang et al., Dense Contrastive Learning for Self-Supervised Visual Pre-Training, CVPR 2021

---

### Official Review · Reviewer_T8CE · 2021-11-08

**Correctness:** 4
**Technical Novelty And Significance:** 2
**Empirical Novelty And Significance:** 3
**Recommendation:** 5
**Confidence:** 4

**Main Review:**

### Pros

- Clear motivation over the prior work and a simple yet effective solution.
- Nice details on the method, e.g., linear scheduling of the acceptance rate, to be hyperparameter-free.
- Nice experimental setup, e.g., comparing with SimCLR and SupCon to provide lower and upper bounds.


### Cons/concerns

1. Limited novelty and inappropriate credit for PCL

The proposed method is somewhat incremental over the prior work, PCL. As PCL develops the core concept of pseudo-label-based contrastive learning, the paper should clearly illustrate the contribution of prior work and its novel innovations. While the paper briefly introduces PCL in the related work section, the paper should clearly state PCL as the preliminary in the main method section.

For example, the "hierarchical semantic definition" part of the paper (page 4) suggests using multiple clusters to consider the different hierarchy levels. However, this technique was already introduced in the original PCL paper (Eq. (11)).


2. Ablation study on the difference from PCL

Putting aside omitting uncertain pseudo-labels, IFND has several minor differences from the PCL. After clearly stating the differences, the paper could provide the ablation study for each modified component, e.g.,
- Confidence of pseudo-label: One may use Eq. (9) of PCL instead of the proposed Eq. (2), although the latter seems to be better as the former does not consider different classes.
- Number of clusters: The original PCL used {25000,50000,100000} but INFD uses {1000,3000,10000}. It can be unfair since the desired number of clusters are 1000 for ImageNet.


3. Comparison with other cluster-based (and other self-supervised) methods

Table 1 suggests that SwAV, another cluster-based method, performs the best. While the paper only verifies the merit over its predecessor, PCL, the empirical contribution could be limited if it underperforms than other self-supervised methods. Thus, the paper should compare IFND with SwAV for all experiments, under the same setup (e.g., 200 epochs).

While BYOL is not cluster-based method, it does not suffer from the suggested false negative issue, which weaken the motivation of this work. The paper could also verify the advantages over the BYOL to strengthen the desirability of proposed method.


Minor comments:
- While the paper introduces two losses: elimination and attraction, the paper concludes that elimination is better for considered scenarios. The paper could mainly introduce the single final method and leave the inferior one for the discussion.
- The definition of false negative $\mathcal{N}(i)$ could be introduced before it is used in Eq. (3).

**Summary Of The Paper:**

The paper proposes incremental false-negative detection (IFND), which has several improvements over prototypical contrastive learning (PCL). The key idea of PCL is to alternatively update (a) the pseudo-labels based on the current encoder and (b) update the encoder based on the current pseudo-labels. However, PCL often converges to the local minima since the bad encoder produces falsy pseudo-labels, further harming the encoder. To tackle the issue, INFD suggests including the confident pseudo-labels while uncertain ones incrementally. IFND consistently improves the PCL in all considered scenarios.

**Summary Of The Review:**

The paper proposes a solid yet somewhat incremental modification from the prior work, PCL. Overall, I think the paper is on the boarderline. I'm willing to raise my score if my concerns: (a) improper credit for PCL, (b) ablation study over PCL, amd (c) comparison with SwAV/BYOL are addressed.

---

> ### Author Response · Authors · 2021-11-23
> **Responses to Reviewer T8CE (2/2)**
>
> > "Comparison with other cluster-based (and other self-supervised) methods"
>
> The SwAV method uses a different learning framework from SimCLR and our approach, and also uses additional training strategies such as multi-crop augmentation. Therefore, it is unfair to directly compare our and the SwAV schemes. To validate the effectiveness of the proposed concept, we apply our incremental false negative removal to the SwAV method. The idea is to prevent that the images with the same semantic concept are assigned with different codes when SwAV computes the codes online. The experiment results are shown in the below table.
>
> | Method  | Top-1 Acc. |
> | :-----: | :-----: |
> | `MoCo v2` | 67.5 |
> | `IFND (MoCo v2)` | 69.7 (2.2%$\uparrow$) |
> | `SwAV` | 72.0 |
> | `IFND (SwAV)` | 72.8 (0.8%$\uparrow$) |
>
> By using either MoCo v2 or SwAV as the baseline learning framework, IFND can consistently boost performance.
> The results demonstrate the advantage of progressively removing the false negatives during contrastive learning.
> More implementation details and discussions are added in Appendix F of the revised manuscript.
>
> The goal of this work is to address the false negative issue for self-supervised contrastive learning. Analyzing the (dis)advantages of contrastive (e.g. SimCLR) and non-contrastive-based (e.g. BYOL) algorithms is not the major focus of this work. Nevertheless, as shown in the above table, the proposed concept improves the SwAV approach, which demonstrates comparable performance to non-contrastive-based learning methods, such as BYOL. Moreover, non-contrastive-based schemes have been shown to have the model collapsing problem in some tasks, such as human pose estimation [1]. In contrast, the proposed approach improves the performance of contrastive learning approaches, which do not suffer from the model collapsing issue.
>
> [1] R. Xie et al., An Empirical Study of the Collapsing Problem in Semi-Supervised 2D Human Pose Estimation, ICCV 2021

---

> ### Author Response · Authors · 2021-11-23
> **Responses to Reviewer T8CE (1/2)**
>
> We thank the reviewer for the insightful comments and suggestions to this work. We address the raised issues below.
>
> ---
>
> > "Limited novelty and inappropriate credit for PCL"
>
> The main contributions of this work consist of 1) highlighting the unfavorable effect of false negatives in self-supervised contrastive learning, 2) introducing a framework that removes the false negatives during contrastive learning, and 3) developing an incremental scheme to obtain more reliable pseudo-labels for detecting false negatives. Our method is significantly different from the PCL approach in several aspects. As described in Section 2, the PCL method uses the instance-level contrastive loss that still suffers from the effect of false negatives. Although our method uses a similar hierarchical semantic concept, it is not our major contribution. We have added the citation to the PCL approach for the clustering strategy and hierarchical semantic definition in Section 3.2 of the revised paper.
>
> ---
>
> > "Ablation study on the difference from PCL"
>
> We present experimental evaluations between our and PCL frameworks in Table 6 of the originally submitted manuscript as below. In the experiments, we first compare $\mathcal{L}_\mathrm{ProtoNCE}$ and $\mathcal{L}_\mathrm{elim}$ to show the effectiveness of removing false negatives. Secondly, we also demonstrate the advantage of using pseudo labels in a linear strategy.
>
> | Method  | Objective | Pseudo Label Assignment | Top-1 Acc. (%) |
> | :-----: | :-----: | :-----: | :-----: |
> | `SimCLR` | $\mathcal{L}_\mathrm{inst}$ | Constant (0%) | 71.5 |
> | `PCL` | $\mathcal{L}_\mathrm{ProtoNCE}$ | Step (0%$\rightarrow$100%) | 72.8 |
> | | $\mathcal{L}_\mathrm{elim}$ | Step (0%$\rightarrow$100%) | 73.1 |
> | | $\mathcal{L}_\mathrm{ProtoNCE}$ | Linear (0%$\rightarrow$100%) | 73.4 |
> | `IFND` | $\mathcal{L}_\mathrm{elim}$ | Linear (0%$\rightarrow$100%) | **74.0** |
>
> As suggested, we conduct the ablation study to understand the impact of the number of clusters. The experimental setup is the same as in Table 6 and Appendix E. The results are shown as follows. The performance of the PCL approach degrades significantly when the number of clusters becomes smaller. In contrast, the proposed method is more robust to the number of clusters.
>
> | Method  | Objective | Numbers of clusters | Top-1 Acc. (%) |
> | :-----: | :-----: | :-----: | :-----: |
> | `PCL` | $\mathcal{L}_\mathrm{ProtoNCE}$ | {2500, 5000, 10000} | 72.8 |
> | | $\mathcal{L}_\mathrm{ProtoNCE}$ | {100, 300, 1000} | 69.9 (2.9$\downarrow$) |
> | | $\mathcal{L}_\mathrm{elim}$ | {2500, 5000, 10000} | 73.2 (0.8$\downarrow$) |
> | `IFND` | $\mathcal{L}_\mathrm{elim}$ | {100, 300, 1000} | 74.0 |
>
> We also conduct the ablation study to compare different strategies to acquire the confidence score and show the results as below. The representation learning with our confidence estimation performs better than the one in PCL. One reason is that the concentration estimation in PCL only assigns a single confidence score for the instances within the same cluster. In contrast, our method respectively computes the confidence score for each instance and also refers to the centroids of other clusters. Hence, it can better estimate the confidence of each individual pseudo label.
>
> | Method  | Objective | Confidence Estimation | Top-1 Acc. (%) |
> | :-----: | :-----: | :-----: | :-----: |
> | `IFND` | $\mathcal{L}_\mathrm{elim}$ | Eq. 2 in our paper | 74.0 |
> | | $\mathcal{L}_\mathrm{elim}$ | Eq. 9 in PCL | 71.2 |
>
> We have added these two studies to Appendix E of the revised manuscript.

---

> ### Comment · Reviewer_T8CE · 2021-11-25
> **Response to the Rebuttal**
>
> I sincerely read all reviews and rebuttals. The rebuttal addressed some of my (and other reviewers') concerns, but it is unsatisfactory considering two weeks of rebuttal periods. The rebuttal was submitted right before the deadline, and thus we had no chance of a second iteration. For this reason, I keep my original rating of weak reject.
>
> I first appreciate that the authors added the detailed ablation study with PCL (concern 2, or simply C2), justifying the design choices of IFND. However, other concerns are not fully resolved, as stated below.
>
> ---
>
> > C1 (also C1 of Reviewer 7ZtC): Novelty over PCL.
>
> I agree that the paper has some contribution over PCL, addressing the false negative issue. However, I respectively disagree that IFND is "significantly different" from PCL.
>
> For example, the rebuttal claims to remove InfoNCE and only use ProtoNCE (original ProtoNCE contains InfoNCE, but I mean the right term of Eq. (11) of PCL) is the major difference over PCL. However, PCL could choose any InfoNCE + ProtoNCE combinations, and they use the InfoNCE term just for an empirical reason. It was a nice suggestion to drop InfoNCE, but I'm not sure it is a significant technical innovation over PCL. On the other hand, while the rebuttal claims IFND uses a "similar" hierarchical semantic concept of PCL, it is the "same" concept in my understanding.
>
> The revised paper added two mentions that IFND components are built on PCL, but they are still under the "proposed method" section.
>
> ---
>
> > C3 (also C1 of Reviewer qg7u): Comparison with non-contrastive methods.
>
> The rebuttal included additional results of SwAV+IFND, which would extend the contribution of the paper. This result is interesting since SwAV is state-of-the-art on various benchmarks (on par with BYOL). Could IFND also gain for the larger setup, e.g., applied on the last row of Table 1? If then, the impact of this paper would be significantly raised. (p.s., while the rebuttal claim that comparison with SwAV is unfair due to the multi-crop, the reader may wonder the final performance of SwAV+multi-crop+IFND) By the way, the current description in Appendix F is somewhat ambiguous. Could you elaborate on the difference from the original SwAV?
>
> ---
>
> > C3 of Reviewer qg7u: Pre-training on multi-object images.
>
> As Reviewer qg7u pointed out, the definition of positive and negative samples are ambiguous for multi-object images, significantly limiting its applicability in real-world situations. It would be appreciated if the rebuttal suggested some ideas to extend IFND for those scenarios.

---

> > ### Author Response · Authors · 2021-11-26
> > **Unsolved concerns from Reviewer T8CE**
> >
> > We thank the reviewer for the additional comments.
> >
> > Sorry for the late responses. Our paper receives a late review (Nov 10) and we need about a week to conduct all experiments and then revise the paper. We would continue to provide more details until the end of Nov 29 to address any concerns. Here are the responses to the reviewer's comments.
> >
> > ---
> >
> > > Novelty over PCL.
> >
> > ProtoNCE is the combination of (1) infoNCE (left term) and (2) prototypical contrastive (right term) losses.
> > In this paper, we provide in-depth analysis on both of them and show the importance to improve both of them:
> >
> > (1) Removing instance-level infoNCE loss requires recognizing that the false negatives have negative effects during contrastive learning.
> > The concept of false negatives is not introduced in PCL. In contrast, we extensively analyze this issue in the paper that leads to the solution of removing infoNCE.
> >
> > (2) Prototypical contrastive loss uses the attraction strategy to handle the pseudo labels (i.e., the right term of Eq. 11 in PCL takes a similar form to Eq. 5 in our paper).
> > In this paper, we provide a detailed comparison between the attraction and suggested elimination strategies using both the theoretical analysis described in Section 3.3 and the quantitative study shown in Table 5.
> > Both studies showed that compared to the elimination scheme, the attraction method is more sensitive to the noisy pseudo labels generated by clustering and performs unfavorably in the unsupervised setting.
> >
> > Overall, both removing false negatives and applying the elimination strategies require meticulous study and algorithmic designs to integrate all modules to achieve state-of-the-art performane. Also, both of the improvements show clear conceptual differences between our and PCL approaches.
> >
> > Clustering and hierarchical semantic concepts are two components of our algorithm. Hence, we include them in Section 3. As suggested, we will revise the title of Section 3 to "Methodology". We certainly agree that we should give due credit to PCL. On the other hand, we believe we have emphasized the differences and importance of the proposed modules to achieve state-of-the-art results.
> >
> > ---
> >
> > > Comparison with non-contrastive methods.
> >
> > We can only train our method on a smaller setup due to limited computing resources and time constraints for the rebuttal. Please note that the performance in Table 8 is the number with multi-crop augmentation (i.e., 2×160 + 4×96 as mentioned in Appendix F).
> >
> > Regarding the difference from the original SwAV, SwAV originally follows the equipartition constraint to compute code online which makes sure the codes for different images in a batch are distinct. To remove the false negatives, we force the codes of semantically similar images to be the same. Specifically, when computing code through Eq. (3), Z originally contains the features directly extracted from different images. In our approach, we use one representative feature to represent the features of semantically similar images by averaging them. Hence, Z includes some same features shared by the semantically similar images. Moreover, during contrastive learning (through minimizing Eq. (2)), we remove the false negatives from the negative sample set (i.e, the dominator).
> >
> > We will add these details to the paper.
> >
> > ---
> >
> > > Pre-training on multi-object images.
> >
> > In [1], the authors define positive/negative samples for multi-object images in a contrastive learning framework. Specifically, they discard the avgpool layer in ResNet and define the "instance" as the feature extracted from a specific region of the image.
> >
> > As such, the false negatives in this work are defined as: the features extracted from the regions which are semantically similar to the anchor region. With this definition, we can extend IFND to multi-object datasets.
> >
> > [1] X. Wang et al.,  Dense Contrastive Learning for Self-Supervised Visual Pre-Training, CVPR 2021

---

> > > ### Comment · Reviewer_T8CE · 2021-11-29
> > > **Response to the 2nd Rebuttal**
> > >
> > > Thank you for the additional response.
> > >
> > > First, I remark that I agree with the proposed two components: (1) removing false negatives and (2) suggestion of elimination strategy are nice improvements over PCL. While I agree with the novelty of the proposed components, I claimed it is *limited* as the overall framework (e.g., clustering and hierarchical concepts) follows the one of PCL.
> > >
> > > I think the newly proposed (yet not fully riped) (1) extension to SwAV (and other non-contrastive methods) and (2) extension to multi-object images would be a clear difference and novelty over PCL. I suggest the revised paper include them in the main text and provide supporting experiments, which would strengthen the impact of the paper rather than being an improvement over PCL.

---

> > > > ### Author Response · Authors · 2021-11-29
> > > > **Responses to Reviewer T8CE**
> > > >
> > > > Thanks for the further response!!
> > > >
> > > > Here, we would like to supplement the experimental results on the multi-object COCO dataset.
> > > > In the experiment, we extend our IFND to dense contrastive learning framework (abbreviated as DenseCL) [1] as mentioned in the previous response.
> > > > Specifically, we perform clustering on the embedding space after the dense projection head, where "instance" is defined as a feature extracted from a specific region in the image.
> > > > In order to remove the false negative samples, we revise the instance-level loss (Eq. (2) in DenseCL) to our elimination loss.
> > > > We follow the training details described in their paper but only pre-train the model in 100 epochs due to time constraints.
> > > > The pre-training model is trained on train2017 and evaluated on val2017 dataset. The results are shown in the below table.
> > > >
> > > > | Method  | AP$^\mathrm{bb}$ | AP$^\mathrm{bb}_\mathrm{50}$ | AP$^\mathrm{bb}_\mathrm{75}$ | AP$^\mathrm{mk}$ | AP$^\mathrm{mk}_\mathrm{50}$ | AP$^\mathrm{mk}_\mathrm{75}$ |
> > > > | :-----: | :-----: | :-----: | :-----: | :-----: | :-----: | :-----: |
> > > > | `Dense CL*` | 38.7 | 58.2 | 42.0 | 34.5 | 55.1 | 37.0 |
> > > > | `IFND (Ours)` | **38.9** | **58.6** | **42.3** | **34.6** | **55.5** | **37.6** |
> > > >
> > > > *We reproduce DenseCL by training 100 epochs for a fair comparison.
> > > >
> > > > The results illustrate that the proposed IFND can also be applied to multi-object datasets.
> > > >
> > > > As suggested, we will (1) move the extension to SwAV to the main paper and (2) provide more implementation details and complete experiment results on the multi-object datasets in the final paper.
> > > >
> > > > [1] X. Wang et al., Dense Contrastive Learning for Self-Supervised Visual Pre-Training, CVPR 2021

---

### Official Review · Reviewer_7ZtC · 2021-11-09

**Correctness:** 3
**Technical Novelty And Significance:** 2
**Empirical Novelty And Significance:** 3
**Recommendation:** 8
**Confidence:** 4

**Main Review:**

Strengths:
1) The paper is easy to follow with clear motivation. The discussion and analysis about “false negatives” on contrastive learning in Fig1 is quite valuable.
2) The authors justify the effectiveness of the proposed method according to the extensive evaluations. Besides, the ablation study also validates the contribution of the main component of the proposed IFND.

Weaknesses:
1) The technical novelty of the proposed method is somewhat marginal since some of its main components (i.e., hierarchical semantic definition) have already been mentioned in PCL (Li et al., 2021). The authors should present more detailed analyses and discussions to highlight the difference between PCL and their proposed IFND. For instance, it will be better if the authors can conduct experiments on the object detection task to demonstrate the consistently superior performance of IFND over PCL.
2) The acceptance rate of the pseudo label is determined without careful consideration. According to Appendix B, the authors empirically set the acceptance rate in a hyperparameter-free manner (i.e., Epoch / Training Epoch). This might make the learning process unstable due to the accumulated pseudo-labeling errors. It might be better if the authors can explore some robust learning techniques (e.g., self-paced learning) to improve learning efficiency.
3) It seems that the experiment results do not validate the effectiveness of the introduced attraction strategy according to Table 5.

**Summary Of The Paper:**

Briefly, this paper presents a simple yet effective method (i.e., IFND) for self-supervised contrastive learning by effectively handling the false-negative samples. Specifically, the authors propose to cluster samples in the embedding space and assign the cluster indices as pseudo labels in an incremental manner during training. The experimental results are extensive and demonstrate the effectiveness of the proposed method.

**Summary Of The Review:**

This paper introduces some interesting insights for contrastive learning. However, its presentation and experiment design could be improved. I will raise my rating if most of my concerns have been well addressed.

---

> ### Author Response · Authors · 2021-11-22
> **Responses to Reviewer 7ZtC**
>
> We thank the reviewer for the insightful and valuable comments. Here are our responses:
>
> ---
>
> > "More detailed analyses and discussions to highlight the difference between PCL and IFND."
>
> The main contributions of this work consist of 1) highlighting the unfavorable effect of false negatives in self-supervised contrastive learning, 2) introducing a framework that removes the false negatives during contrastive learning, and 3) developing an incremental scheme to obtain more reliable pseudo-labels for detecting false negatives. Our method is significantly different from the PCL approach in several aspects. As described in Section 2, PCL adopts the instance-level contrastive loss that still suffers from the effect of false negatives. Although our method uses a similar hierarchical semantic concept, it is not our major contribution.
>
> Moreover, we present in-depth experimental evaluations between our and PCL frameworks in Table 6 of the originally submitted manuscript as below. In the experiments, we first compare $\mathcal{L}_\mathrm{ProtoNCE}$ and $\mathcal{L}_\mathrm{elim}$ to show the effectiveness of removing false negatives. Secondly, we also demonstrate the advantage of using pseudo labels in a linear strategy.
>
> | Method  | Objective | Pseudo Label Assignment | Top-1 Acc. (%) |
> | :-----: | :-----: | :-----: | :-----: |
> | `SimCLR` | $\mathcal{L}_\mathrm{inst}$ | Constant (0%) | 71.5 |
> | `PCL` | $\mathcal{L}_\mathrm{ProtoNCE}$ | Step (0%$\rightarrow$100%) | 72.8 |
> | | $\mathcal{L}_\mathrm{elim}$ | Step (0%$\rightarrow$100%) | 73.1 |
> | | $\mathcal{L}_\mathrm{ProtoNCE}$ | Linear (0%$\rightarrow$100%) | 73.4 |
> | `IFND` | $\mathcal{L}_\mathrm{elim}$ | Linear (0%$\rightarrow$100%) | **74.0** |
>
> As suggested, we also provide more experimental results on the object detection and semantic segmentation tasks using the COCO dataset. The results are shown in the below table and have been added to Section 4.2 in the revised paper (revised text is marked in red).
>
> | Method  | AP$^\mathrm{bb}$ | AP$^\mathrm{bb}_\mathrm{50}$ | AP$^\mathrm{bb}_\mathrm{75}$ | AP$^\mathrm{mk}$ | AP$^\mathrm{mk}_\mathrm{50}$ | AP$^\mathrm{mk}_\mathrm{75}$ |
> | :-----: | :-----: | :-----: | :-----: | :-----: | :-----: | :-----: |
> | `Supervise` | 40.0 | 59.9 | 43.1 | 34.7 | 56.5 | 36.9 |
> | `MoCo` | 40.7 | 60.5 | 44.1 | 35.4 | 57.3 | 37.6 |
> | `PCL` | 41.0 | 60.8 | 44.2 | 35.6 | 57.4 | 37.8 |
> | `IFND (Ours)` | **41.8** | **61.2** | **44.5** | **36.1** | **57.6** | **38.5** |
>
> Overall, the proposed IFND approach performs favorably against the PCL and supervised pre-training schemes on these two tasks.
>
> ---
>
> > "The acceptance rate of the pseudo label is determined without careful consideration."
>
> The main idea of this work is to progressively remove false negatives for self-supervised contrastive learning. We find that even the simple linear assignment strategy can validate our concept and improve the performance as shown in Table 5. We appreciate the reviewer's suggestion and plan to explore more schemes to determine the acceptance rate in our future work.
>
> ---
>
> > "The experiment results do not validate the effectiveness of the introduced attraction strategy"
>
> In Section 3.3, we compare two options to remove the false negatives from the contrastive learning objective: attraction and elimination. We give both a theoretical analysis in Section 3.3 and a quantitative study in Table 5 to discuss and compare their behavior and demonstrate that the elimination strategy is more suitable for leveraging the false negatives detected by our approach. As a result, we use the elimination strategy in all other experiments.

---

> > ### Comment · Reviewer_7ZtC · 2021-11-29
> > **Response to the Rebuttal**
> >
> > Thanks a lot for the updated results and explanations.
> > Most of my concerns have been well addressed. I will give a higher rating.

---

### Author Response · Authors · 2021-11-29
**Summary of the general concerns and author responses**

Near the end of the discussion, we thank the reviewer again for giving lots of insightful and valuable suggestions. Here, we summarize the general concerns raised by the reviewers and our corresponding responses as below.
We will carefully make sure all responses mentioned below will be added to the final paper to fully solve the concerns.

---

> Reviewers 7ZtC, T8CE: "Novelty and credit to PCL"

+ We highlight three differences from PCL: 1) removing the false negatives, 2) applying the elimination strategy, and 3) developing an incremental scheme to use pseudo labels.

+ We provide several studies to emphasize the importance to improve each proposed component. For 1), we explore the effect of false negatives for self-supervised contrastive learning in Section 4.3 and Table 5. For 2), we present the theoretical and quantitative comparison between the suggested elimination and the attraction (which is applied in PCL) strategies in Section 3.3 and Table 6. For 3), we show that following the training, the embedding space progressively becomes more semantically structural in Section 4.5 and Appendix G. These studies explain why the proposed IFND outperforms PCL with a clear gap and also achieves a state-of-the-art result.

+ Thanks to the suggestion from reviewers T8CE and 7ZtC. We also provide an in-depth comparison with PCL in terms of 1) objective function, 2)  pseudo label assignment, 3) numbers of clusters, and 4) confidence estimation in Appendix E of the revised paper.
The comparison not only shows the superiority of the proposed concepts but also illustrates that IFND is more robust to the number of clusters, which is important for an unsupervised learning framework.

+ Regarding the concepts of clustering and hierarchical semantic definition, we have added the citation of PCL in Section 3.2. We will move the part of the hierarchical semantic definition to Section 4.1 Implementation Details, and revise the title of Section 3 to "Methodology" in the final paper.

---

> Reviewers T8CE, qg7u: "Extension of IFND to other methods and multi-object datasets"

+ Thanks to the suggestion from reviewers T8CE and qg7u. In Appendix F of the revised paper, we have added the extension of IFND which uses SwAV as a baseline framework. We demonstrate a 0.8% performance gain from IFND compared to vanilla SwAV. We will move the result to Section 4 in the main paper and add more implementation details in the final paper.

+ As suggested by reviewers qg7u and T8CE, we also extend IFND to the multi-object datasets by using dense contrastive learning [1] as a baseline framework. We show the potential of IFND on the multi-object images as in the supplemented experiment. We will provide the complete experiment results in the final paper.

[1] X. Wang et al., Dense Contrastive Learning for Self-Supervised Visual Pre-Training, CVPR 2021

---

> Reviewers 7ZtC, qg7u: "Evaluation on object-detection tasks"

+ We have added the experiments in Section 4.2 of the revised paper. The results show that the proposed IFND approach performs favorably against the PCL and supervised pre-training schemes on both object-detection and semantic segmentation.

We believe the corresponding responses and results have covered most of your concerns.

Please let us know whether you have additional questions or not.

Thanks,

---

### Decision · Program_Chairs · 2022-01-20

**Decision:**

Accept (Poster)

**Comment:**

While the reviewers were somewhat split on this paper, they all found some strengths, and pointed out some weaknesses. Among these the main seems to be the somewhat incremental nature of the work, in particular with respect to PCL. As the authors point out, the differences w.r.t. PCL are meaningful and include the main thrust of the paper (removal of false negatives), and the results do indicate usefulness of the proposed approach. Given the wide interest in self-supervision I think the paper is above bar for acceptance.